# Predicting early cessation of exclusive breastfeeding using machine learning techniques

**Freja Marie Nejsum**[1]*, **Rikke Wiingreen**[1,2], **Andreas Kryger Jensen**[3,4], **Ellen Christine Leth Løkkegaard**[2,5], **Bo Mølholm Hansen**[1,2]

1 Department of Pediatrics, Copenhagen University Hospital—North Zealand, Hillerød, Denmark, 2 Department of Clinical Medicine, University of Copenhagen, Copenhagen, Denmark, 3 Department of Public Health, University of Copenhagen, Copenhagen, Denmark, 4 Department of Clinical Research, Copenhagen University Hospital—North Zealand, Hillerød, Denmark, 5 Department of Gynaecology and Obstetrics, Copenhagen University Hospital—North Zealand, Hillerød, Denmark

* frejanejsum@gmail.com

## Abstract

### Background

Identification of mother-infant pairs predisposed to early cessation of exclusive breastfeeding is important for delivering targeted support. Machine learning techniques enable development of transparent prediction models that enhance clinical applicability. We aimed to develop and validate two models to predict cessation of exclusive breastfeeding within one month among infants born after 35 weeks gestation using machine learning techniques.

### Methods

Utilizing a nationwide dataset from Statistics Denmark, including infants born between the 1st of January 2014 and the 31st of December 2015, we employed random forest machine learning to develop two predictive models. The first model included 11 well-established factors associated with cessation of exclusive breastfeeding within one month. The second model was expanded to include 21 additional factors associated with complications during pregnancy and delivery that potentially impede breastfeeding. Feature importance was applied to elucidate the factors driving model predictions.

### Results

The dataset comprised 110,206 infants and 106,835 mothers. The first model predicted cessation of exclusive breastfeeding within one month with an area under the receiver operating curve of 62.0% (95% confidence interval 61.3% - 62.7%) and an accuracy of 60.4% (95% confidence interval 59.8% - 61.0%). The second model predicted cessation of exclusive breastfeeding within one month with an area under the receiver operating curve of 62.2% (95% confidence interval 61.5% - 62.9%) and an accuracy of 60.0% (95% confidence interval 59.3% - 60.6%). In both models, birthplace, maternal education, delivery mode, and

**Data Availability Statement:** Data cannot be shared publicly because of Danish law and protection of patient privacy. Data are available through secure online access to Statistics

Denmark. Further information regarding data access can be found on Statistics Denmark's website http://dst.dk/en/ TilSalg/Forskningsservice or by contacting Statistics Denmark on e-mail forskningsservice@dst.dk or phone +4539173130. We cannot guarantee access to data, but we will gladly assist interested institutions upon any requests.

**Funding:** The author(s) received no specific funding for this work.

**Competing interests:** The authors have declared that no competing interests exist.

maternal body mass index were the most important factors influencing the overall model performance.

## Conclusions

The two models could not accurately predict cessation of exclusive breastfeeding within one month among infants born after 35 weeks gestation. Contrary to our expectations, including additional factors in the model did not increase model performance.

## Introduction

Breastfeeding confers multiple health benefits for infants and mothers that extend beyond solely the neonatal period [1, 2]. In accordance, the World Health Organization recommends exclusive breastfeeding for the first six months after birth [3]. In Denmark, the preponderance of expectant mothers expresses an intention to breastfeed [4]. Despite these intentions, 40% of mothers initiating breastfeeding encounter early breastfeeding problems [5]. Thus, the adherence to the World Health Organization's recommended six-month exclusive breastfeeding duration is low, with approximately 11% compliance in Denmark [6–8].

The first step to achieve exclusive breastfeeding for six months after birth is to establish exclusive breastfeeding. Breastfeeding establishment is a multifaceted process. Many mother-infant pairs establish breastfeeding without any complications. Existing evidence consistently affirms the influence of several factors, including maternal age, maternal smoking, maternal body mass index, socioeconomic status, parity, delivery mode, infant sex, gestational age, and being small-for-gestational-age [9–13]. A recent, nationwide cohort study, encompassing more than 100,000 infants, reaffirms the significance of these associations [14]. Nevertheless, it remains plausible that additional unexplored factors also exert an influence on breastfeeding establishment. Complications during pregnancy and delivery can disrupt crucial practices that are important for breastfeeding establishment e.g., immediate skin-to-skin contact and early initiation of breastfeeding [15]. Many mothers experience complications during pregnancy and delivery and, despite compelling physiological rationales, the influence of such complications on breastfeeding establishment remains insufficiently explored.

Several studies have attempted to develop prediction models aimed at enhancing breast-feeding interventions [16–20]. Nevertheless, a validated model to accurately predict well-established breastfeeding that persists beyond hospital discharge remains absent. Existing models include limited predictors and are constructed using relatively small datasets. Many breast-feeding problems can be remedied by timely support [15] and this underscores the need for further research aimed at targeting quality breastfeeding interventions.

In recent decades, machine learning has increasingly been used in prediction models and has the potential to increase model performance [21]. Prediction models can be developed using machine learning techniques. Machine learning is a subset of artificial intelligence that employs a data-driven approach to model development [21, 22]. It is increasingly applied in various medical fields [22]. A previous study found that machine learning techniques produced more accurate model predictions of in-hospital breastfeeding compared to traditional statistics [16]. Recent advances in explainable artificial intelligence enable transparent explanations of model predictions that enhance clinical applicability [23].With this study, we aimed to develop and validate two models to predict cessation of exclusive breastfeeding within one month among infants born after 35 weeks gestation using machine learning techniques, with

potential for application in the hospital immediately after birth to target support interventions. We hypothesized that including additional predictors in the model would produce more accurate predictions.

## Methods

Using a retrospective nationwide cohort of infants born in Denmark between the 1st of January 2014 and the 31st of December 2015, we developed and validated two models predicting cessation of exclusive breastfeeding within one month. The cohort has been thoroughly characterized in a previous study [14].

### Data source

Our dataset was obtained from multiple nationwide registers held by Statistics Denmark and The Danish Health Data Authority including The Danish National Child Health Register, The Danish National Patient Register, The Danish Medical Birth Register, The Danish Education Registers, The Danish Register of Causes of Death, and The Danish Civil Registration System. In Denmark, all individuals are assigned distinctive Central Personal Register numbers upon birth or immigration, which enables consistent linkage of data across the registers [24].

### Participants

The study population included mother-infant pairs born in Denmark between the 1st of January 2014 and the 31st of December 2015. Infants meeting the following criteria were excluded from the study population: Gestational age below 35 weeks and 0 days, missing data on gestational age or birthweight, gestational age or birth weight outliers, and death or migration of the infant or mother within the first month after birth. The 35 weeks cutoff was chosen because, in Denmark, most infants born at gestational ages below 35 weeks and 0 days routinely are admitted to neonatal wards where they receive additional support to establish breastfeeding. Outliers were excluded under the presumption that they stemmed from errors in data coding. Gestational age outliers were defined as gestational ages at birth above 44 weeks and 0 days. Birth weight outliers were defined as birth weights deviating more than five standard deviations from the mean of the study population calculated as described by Marsál et al. [25].

### Outcome

The outcome was cessation of exclusive breastfeeding within one month, as exclusive breastfeeding usually is well-established at this point and infants born at gestational ages above 34 weeks and 6 days routinely are discharged from the hospital beforehand.

Data on cessation of exclusive breastfeeding were retrieved from The Danish National Child Register [26]. In the Danish National Child Register, exclusive breastfeeding is defined as feeding the infant solely with breast milk except for water and maximum one formula feeding per week after hospital discharge as described by The Danish Health Authority [27]. Thus, this definition was applied in the current study. It is an adaption of The World Health Organization's definition of exclusive breastfeeding to suffice Danish conditions [28].

In Denmark, health visitors routinely conduct free home visits during the infant's first year, with over 95% of parents utilizing the services [8]. In the first month, health visitors conduct minimum one home visit in the first week and one home visit between the second and fourth week. It is possible to receive extra visits. During these visits, the health visitors collect information on the date of exclusive breastfeeding cessation and subsequently report it to The Danish National Child Register. This practice has been mandatory since 2011 but data are only

considered complete from the 1st of January 2014 [29]. The reporting is conducted via municipalities, which are local administrative divisions responsible for public services within specific geographic areas of Denmark. This leads to considerable delay in reporting of data to The Danish National Child Register. Further, post-registrations dating several years back are possible, thus data on can only be considered complete after years [26].

Infants, who did not initiate exclusive breastfeeding, were not registered with a record on cessation of exclusive breastfeeding in The Danish National Child Health Register [29]. Consequently, infants without a record on cessation of exclusive breastfeeding in the register were classified as having ceased exclusive breastfeeding within the first month after birth.

## Predictors

We developed two models to predict cessation of exclusive breastfeeding within one month. The first model included 11 well-established risk factors for ceasing exclusive breastfeeding within one month: Maternal age, maternal pre-pregnancy body mass index, maternal smoking, maternal education, birthplace, parity, multiple birth, delivery mode, infant's sex, gestational age, and birthweight [9–14]. Maternal education was considered an indicator of socioeconomic status and divided into four levels. Level one (lowest) comprising International Standard Classification of Education 2011 (ISCED) level 1–2, level two comprising ISCED level 3, level three comprising ISCED level 5–6, and level four (highest) comprising ISCED level 7–8 [30]. Birthplace was divided into five regions (Region A-E) corresponding to the healthcare regions of Denmark. Delivery mode was divided into vaginal delivery and cesarean section, with the latter further stratified into emergency and elective cesarean section. Gestational age was defined as completed weeks. In Denmark, gestational age is typically determined through first-trimester ultrasonography performed in approximately 92% of pregnancies [31]. Birth weight deviation was calculated as described by Marsál et al. [25] and divided into three levels: Small-for-gestational-age (below -2 standard deviations from the reference mean), appropriate-for-gestational-age (-2 to 2 standard deviations from the reference mean), and large-for-gestational-age (above 2 standard deviations from the reference mean).

The second model was expanded to include 21 additional factors. In addition to the factors included in model 1, model 2 further incorporated: Ethnicity, maternal psychiatric disease, maternal somatic chronic disease, preeclampsia and eclampsia, hemorrhage in early pregnancy (before gestational age 12 weeks and 0 days), gestational diabetes mellitus, liver disease, hemorrhage in late pregnancy (after gestational age 12 weeks and 0 days), preterm premature rupture of membranes, placenta previa, abruptio placenta, abnormal forces of labor, uterine rupture, postpartum hemorrhage, retention of placenta or membranes, perineal tear, labor induction, forceps or vacuum extraction, regional anesthesia, general anesthesia, and Apgar score at five minutes after birth. The International Classification of Diseases 10th revision (ICD-10) and NOMESCO Classification of Surgical Procedure codes used to define maternal psychiatric disease (within two years preceding birth), maternal somatic chronic disease (within ten years preceding birth), and the factors associated with complications during pregnancy and delivery can be found in S1–S3 Tables.

## Missing data

Missing data were handled using multiple imputation. We generated ten imputed datasets using the R-package 'mice' [32]. Numeric variables were imputed using predictive mean matching, unordered categorical variables were imputed using logistic regression, and ordered categorical variables were imputed using proportional odds [32].

## Statistical analysis methods

Statistical analyses were made on each of the ten imputed datasets and subsequently combined into one estimate using Rubin's rule [33].

**Data allocation for model development and validation.** The data were divided into one dataset for model development and one dataset for model validation based on the infants' birth month to prevent bias from time-related changes. The dataset for model development comprised all infants born in January, February, March, May, June, July, September, October, and November. The dataset for model validation comprised all infants born in April, August, and December.

**Model development.** To build the two prediction models, we employed Breiman's random forest algorithm using the R-package 'randomForest' [34]. The random forest algorithm is a machine learning technique. It uses bootstrapped samples to construct multiple decision trees, selecting a subset of variables as potential predictors at each split. To tune the models, we adjusted the number of variables sampled at each split and the number of trees to grow in order to minimizing the out-of-bag-error. The number of variables sampled at each split was set to two and the number of trees was set to 500.

The models were trained on the dataset for model development. To train the two models, we applied ten-fold cross validation using the R-package 'caret' [35]. In ten-fold cross validation, the dataset is divided into ten equally sized folds. The model is iteratively trained and tested ten times. During each iteration, one distinct fold is used as the test set, while the remaining nine folds serve as the training set. This ensures that each data point is used for both training and testing, thereby enhancing the precision of the performance estimation [36]. We employed multiple metrics to evaluate the performance of the models including the area under the receiver operating curve (AUC), the area under the precision-recall curve, accuracy, sensitivity, specificity, positive predictive value, negative predictive value, and the Brier score.

**Model validation.** The performance of the two models were evaluated using the dataset for model validation. We employed multiple metrics to evaluate the performance of the models including the AUC, the area under the precision-recall curve, accuracy, sensitivity, specificity, positive predictive value, negative predictive value, and the Brier score.

**Feature importance analysis.** We employed feature importance analysis to gain insight into the models' prediction making processes, which enhances clinical applicability [23]. Feature importance analysis identifies the most important predictors for overall model performance. We used the R-package 'randomForest' to calculate feature importance (based on mean decrease in accuracy) [32]. To assess the stability of the feature importance analyses, we employed the method of sequential rank agreement as described by Ekstrøm et al. [37] using the R-package 'SuperRanker'. All analyses were conducted using R-version 4.2.1 [38].

## Ethics

In accordance with the General Data Protection Regulation, the study was approved by the data responsible institute (Capital Region of Denmark—Approval number P-2019-280). In Denmark, register-based studies conducted for scientific purposes do not require informed consent from individual study participants or further ethical approvals.

The study was reported in compliance with the guidelines outlined in Strengthening the Reporting of Observational Studies in Epidemiology (STROBE) [39] and Transparent Reporting of a Multivariable Prediction Model for Individual Prognosis or Diagnosis (TRIPOD) [40].

## Results

The Danish Medical Birth Register included records on 116,585 infants born between the 1st of January 2014 and the 31st of December 2015. After exclusion of infants with gestational age below 35 weeks and 0 days (3,149; 2.7%), missing perinatal data (3,008; 2.6%), and death or migration within one month of birth (222; 0.2%), the study population comprised 110,206 infants and their 106,835 mothers. This corresponds to 94.5% of the Danish birth cohort in the two-year period (Fig 1).

In the study population, 48,643 infants (44,1%) ceased exclusive breastfeeding within one month. This group included 31,857 infants (28,9%) who did not initiate exclusive breastfeeding and 16,786 infants (15,2%) who began exclusively breastfeeding but discontinued within the first month of birth.

Table 1 shows the baseline characteristics of the study population. In the study population, 2,810/110,206 (2,5%) had missing data on minimum one of the predictors in model 1, while 9,481/110,206 (8,6%) had missing data on minimum one of the predictors in model 2.

The dataset for model development comprised 83,385 infants, while the dataset for model validation comprised 26,821 infants.

Table 2 shows the models' prediction of exclusive breastfeeding in the dataset for model validation.

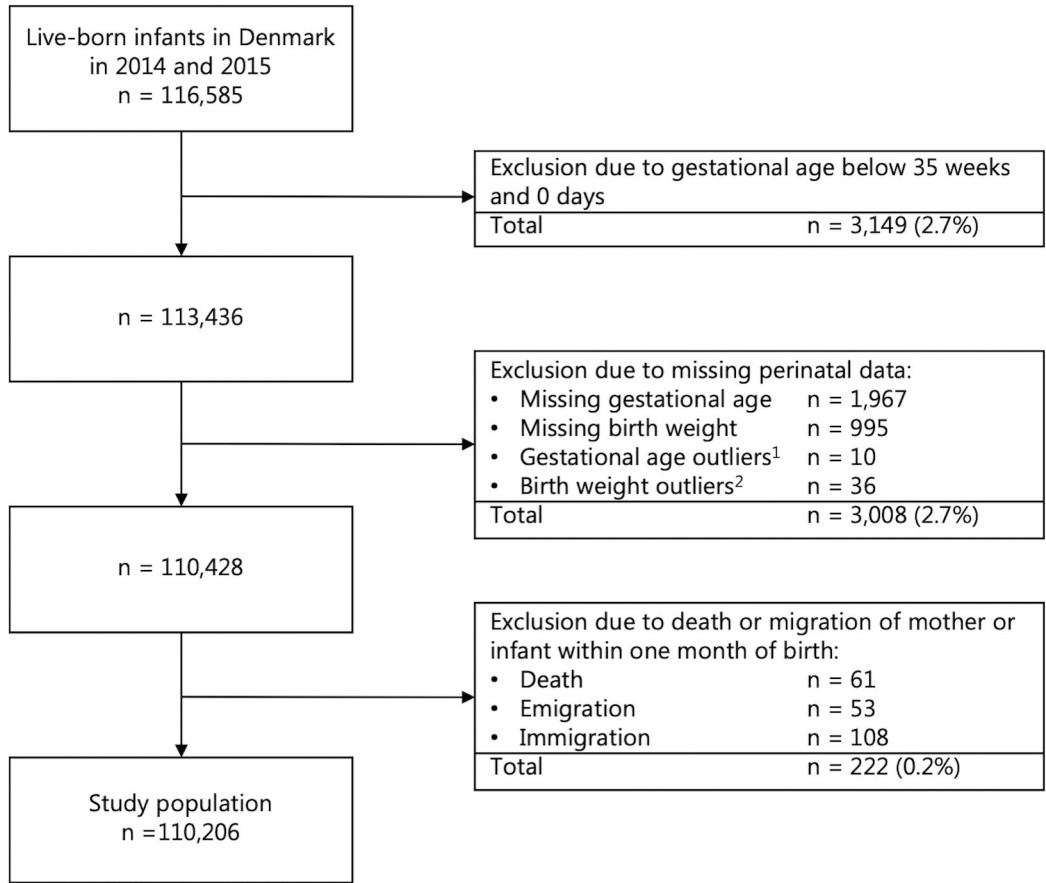

**Fig 1. The study population of infants born in 2014 and 2015 derived from registers held by Statistics Denmark.**
[1]Gestational ages at birth above 44 weeks and 0 days. [2]Birth weights deviating more than five standard deviations from the mean of the study population.

**Table 1. Baseline characteristics of the study population of infants born in 2014 and 2015 based on data from registers held by Statistics Denmark.**

| Characteristic | Total | Exclusive breastfeeding cessation within one month | |
|---|---|---|---|
| | | No | Yes |
| | n = 110,206 | n = 61,563 | n = 48,643 |
| Maternal age, mean (SD) | 30.4 (5.0) | 30.5 (4.9) | 30.3 (5.2) |
| Maternal body mass index, mean (SD) | 24.4 (6.9) | 23.9 (6.5) | 25.0 (7.3) |
| • Missing | 826 | 436 | 390 |
| Maternal smoking, n (%) | | | |
| • Yes | 12,040 (10.9) | 5,505 (8.9) | 6,535 (13.4) |
| • Missing | 817 (0.7) | 446 (0.7) | 371 (0.8) |
| Maternal education[1], n (%) | | | |
| • Level one (lowest) | 13,396 (12.2) | 5,987 (9.7) | 7,409 (15.2) |
| • Level two | 33,443 (30.3) | 17,173 (27.9) | 16,270 (33.4) |
| • Level three | 38,675 (35.1) | 22,466 (36.5) | 16,209 (33.3) |
| • Level four (highest) | 23,861 (21.7) | 15,575 (25.3) | 8,286 (17.0) |
| • Missing | 831 (0.8) | 362 (0.6) | 469 (1.0) |
| Maternal ethnicity, n (%) | | | |
| • Danish | 87,829 (79.7) | 49,333 (80.1) | 38,496 (79.1) |
| • Immigrants | 19,150 (17.4) | 10,517 (17.1) | 8,633 (17.7) |
| • Descendants | 3,188 (2.9) | 1,703 (2.8) | 1,485 (3.1) |
| • Missing | 39 (0.0) | 10 (0.0) | 29 (0.1) |
| Maternal chronic somatic disease[2], n (%) | | | |
| • Yes | 14,384 (13.1) | 7,781 (12.6) | 6,603 (13.6) |
| Maternal psychiatric disease[3], n (%) | | | |
| • Yes | 7,587 (6.9) | 3,667 (6.0) | 3,920 (8.1) |
| Parity, n (%) | | | |
| • Primiparous | 51,718 (46.9) | 29,441 (47.8) | 22,277 (45.8) |
| Birthplace, n (%) | | | |
| • Region A | 40,276 (36.5) | 25,809 (41.9) | 14,467 (29.7) |
| • Region B | 12,213 (11.1) | 6,463 (10.5) | 5,750 (11.8) |
| • Region C | 21,227 (19.3) | 10,144 (16.5) | 11,083 (22.8) |
| • Region D | 26,494 (24.0) | 14,977 (24.3) | 11,517 (23.7) |
| • Region E | 9,588 (8.7) | 3,948 (6.4) | 5,640 (11.6) |
| • Missing | 408 (0.4) | 222 (0.4) | 186 (0.4) |
| Preeclampsia and eclampsia, n (%) | | | |
| • Yes | 3,890 (3.5) | 1,951 (3.2) | 1,939 (4.0) |
| Hemorrhage in early pregnancy, n (%) | | | |
| • Yes | 4,683 (4.2) | 2,531 (4.1) | 2,152 (4.4) |
| Gestational diabetes mellitus, n (%) | | | |
| • Yes | 4,138 (3.8) | 1,988 (3.2) | 2,150 (4.4) |
| Liver disorders, n (%) | | | |
| • Yes | 1,555 (1.4) | 850 (1.4) | 705 (1.4) |
| Preterm premature rupture of membranes, n (%) | | | |
| • Yes | 3,319 (3.0) | 1,699 (2.8) | 1,620 (3.3) |
| Placenta previa, n (%) | | | |
| • Yes | 708 (0.6) | 381 (0.6) | 327 (0.7) |
| Abruptio placentae, n (%) | | | |
| • Yes | 777 (0.7) | 433 (0.7) | 344 (0.7) |
| Antepartum hemorrhage, n (%) | | | |

*(Continued)*

**Table 1.** (*Continued*)

| Characteristic | Total | Exclusive breastfeeding cessation within one month | |
|---|---|---|---|
| | | No | Yes |
| | n = 110,206 | n = 61,563 | n = 48,643 |
| • Yes | 2,237 (2.0) | 1,194 (1.9) | 1,043 (2.1) |
| Abnormalities of forces of labour, n (%) | | | |
| • Yes | 21,776 (19.8) | 12,329 (20.0) | 9,447 (19.4) |
| Uterine rupture, n (%) | | | |
| • Yes | 176 (0.2) | 76 (0.1) | 100 (0.2) |
| Perineal tear[4], n (%) | | | |
| • Yes | 2,218 (2.0) | 1,242 (2.0) | 976 (2.0) |
| Postpartum hemorrhage, n (%) | | | |
| • < 500 mL | 79,766 (72.4) | 45,611 (74.1) | 34,155 (70.2) |
| • 500–999 mL | 17,468 (15.9) | 9,315 (15.1) | 8,153 (16.8) |
| • > 999 mL | 6,550 (5.9) | 3,513 (5.7) | 3,037 (6.2) |
| • Missing | 6,422 (5.8) | 3,124 (5.1) | 3,298 (6.8) |
| Retained placenta and membranes, n (%) | | | |
| • Yes | 1,896 (1.7) | 1,093 (1.8) | 803 (1.7) |
| Labor induction, n (%) | | | |
| • Medical | 15,198 (13.8) | 8,195 (13.3) | 7,003 (14.4) |
| • Mechanical | 6,763 (6.1) | 3,659 (5.9) | 3,104 (6.4) |
| • Both | 3,926 (3.6) | 1,903 (3.1) | 2,023 (4.2) |
| Regional anesthesia during labor or cesarean section, n (%) | | | |
| • Yes | 41,138 (37.3) | 21,280 (34.6) | 19,858 (40.8) |
| General anesthesia, n (%) | | | |
| • Yes | 2,595 (2.4) | 1,241 (2.0) | 1,354 (2.8) |
| Cesarean section, n (%) | | | |
| • Emergency | 12,407 (11.3) | 6,201 (10.1) | 6,206 (12.8) |
| • Elective | 10,529 (9.6) | 4,929 (8.0) | 5,600 (11.5) |
| • Missing | 110 (0.1) | 63 (0.1) | 56 (0.1) |
| Forceps or vacuum delivery, n (%) | | | |
| • Yes | 7,500 (6.8) | 4,418 (7.2) | 3,082 (6.3) |
| Multiple birth, n (%) | | | |
| • Yes | 3,205 (2.9) | 1,181 (1.9) | 2,024 (4.2) |
| Sex, n (%) | | | |
| • Female | 53,726 (48.8) | 30,231 (49.1) | 23,495 (48.3) |
| Gestational age, n (%) | | | |
| • 35 weeks | 1,449 (1.3) | 755 (1.2) | 694 (1.4) |
| • 36 weeks | 2,592 (2.4) | 1,214 (2.0) | 1,378 (2.8) |
| • 37 weeks | 5,955 (5.4) | 2,838 (4.6) | 3,117 (6.4) |
| • 38 weeks | 15,490 (14.1) | 8,027 (13.0) | 7,463 (15.3) |
| • 39 weeks | 24,357 (22.1) | 13,706 (22.3) | 10,651 (21.9) |
| • 40 weeks | 32,174 (29.2) | 18,670 (30.3) | 13,504 (27.8) |
| • 41 weeks | 25,901 (23.5) | 14,997 (24.4) | 10,904 (22.4) |
| • ≥ 42 weeks | 2,290 (2.0) | 1,356 (2.2) | 932 (1.9) |
| Birth weight, mean (SD) | 3,510 (512) | 3,520 (499) | 3,490 (527) |
| Birth weight deviation, n (%) | | | |
| • Small for gestational age | 2,805 (2.5) | 1,329 (2.2) | 1,476 (3.0) |
| • Appropriate for gestational age | 104,403 (94.7) | 58,708 (95.4) | 45,695 (93.9) |

(*Continued*)

**Table 1.** (Continued)

| Characteristic | Total | Exclusive breastfeeding cessation within one month | |
|---|---|---|---|
| | | No | Yes |
| | n = 110,206 | n = 61,563 | n = 48,643 |
| • Large for gestational age | 2,998 (2.7) | 1,526 (2.5) | 1,472 (3.0) |
| Apgar score at five minutes, n (%) | | | |
| • < 7 | 659 (0.6) | 331 (0.5) | 328 (0.7) |
| • Missing | 475 (0.4) | 258 (0.4) | 217 (0.4) |

[1]Maternal education: Level one = International Standard Classification of Education 2011 (ISCED) 1–2, level two = ISCED 3, level three = ISCED 5–6, level four = ISCED 7–8.

[2]Maternal chronic somatic disease within ten years before birth

[3]Maternal psychiatric disease within two years before birth

[4]Third and fourth degree perineal tear

Fig 2 shows the receiver operating curves for model 1 and model 2. The two receiver operating curves were almost identical, with an AUC on 62.9% for model 1 and 62.8% for model 2 (Fig 2, Table 3).

Table 3 shows the performance metrics for model 1 and model 2 in the dataset for model validation. The models performed similar across all performance metrics except sensitivity and specificity. The sensitivity of model 1 was 34.6%, whereas the sensitivity of model 2 was 25.5%. The specificity of model 1 was 80.5%, whereas the specificity of model 2 was 86.9%.

Fig 3 shows the feature importance plot for model 1. The sequential rank agreement was stable across the four highest ranking predictors (S1 Fig), which were birthplace, maternal education, maternal body mass index, and cesarean section.

Fig 4 shows the feature importance plot for model 2. The sequential rank agreement was stable across the eleven highest ranking predictors (S2 Fig), which were birthplace, maternal education, maternal body mass index, cesarean section, maternal smoking, regional anesthesia, maternal age, multiple birth, gestational age, labor induction, and parity.

## Discussion

We used machine learning techniques to develop and validate two models to predict cessation of exclusive breastfeeding within one month using a nationwide cohort including more than 110,000 infants. The two models exhibited a relatively low accuracy in prediction of exclusive breastfeeding cessation within one month. Intriguingly, the inclusion of 21 additional factors in the second model did not result in improved predictive performance.

**Table 2. Confusion matrix for the models' prediction of exclusive breastfeeding cessation within one month in the dataset for model validation.**

| | | Model 1 | | Model 2 | |
|---|---|---|---|---|---|
| | | Actual exclusive breastfeeding cessation within one month | | Actual exclusive breastfeeding cessation within one month | |
| | | Yes | No | Yes | No |
| Predicted exclusive breastfeeding cessation within one month | Yes | 4,073/11,762 | 2,934/15,059 | 3,003/11,762 | 1,971/15,059 |
| | No | 7,689/11,762 | 12,125/15,059 | 8,759/11,762 | 13,088/15,059 |

Model 1 includes 11 maternal and perinatal characteristics.

Model 2 includes 32 maternal, obstetrical, and perinatal characteristics.

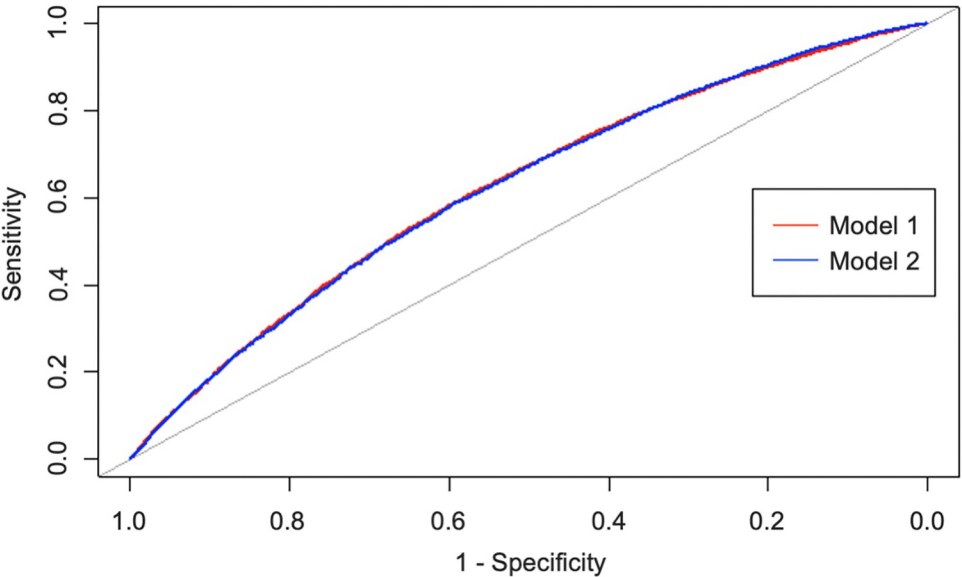

**Fig 2. The receiver operating curves for the models' prediction of exclusive breastfeeding cessation within one month in the dataset for model validation.** Model 1 includes 11 maternal and perinatal characteristics. Model 2 includes 32 maternal, obstetrical, and perinatal characteristics.

The register-based design presents some general advantages including prospective data collection in a real-world setting [41]. The quality of breastfeeding data is especially susceptible to selection, self-reporting, and recall bias [42, 43]. In the present study, selection bias was minimized by the nationwide study population and ability to handle censoring using the unique Central Personal Registration number. The use of routinely collected exclusive breastfeeding data by health visitors contributes to diminishing self-reporting and recall bias.

We applied random forest machine learning to build the prediction models, which holds several advantages compared to traditional statistics including the ability to take non-linearity and interaction into account [36]. We chose the random forest algorithm because of its widespread acceptance as a traditional machine learning method, recognized for its flexibility and

**Table 3. Performance of the models' prediction of exclusive breastfeeding cessation within one month in the dataset for model validation.**

| Performance metric, % (95% CI) | Model 1 | Model 2 |
|---|---|---|
| | **n = 26,821** | **n = 26,821** |
| Area under the receiver operating curve | 62.0 (61.3–62.7) | 62.2 (61.5–62.9) |
| Area under the precision recall curve | 47.8 (46.9–48.7) | 47.5 (46.8–48.2) |
| Accuracy | 60.4 (59.8–61.0) | 60.0 (59.3–60.6) |
| Sensitivity | 34.6 (33.8–35.5) | 25.5 (24.7–26.3) |
| Specificity | 80.5 (79.9–81.1) | 86.9 (86.4–87.4) |
| Positive predictive value | 58.1 (57.0–59.3) | 60.4 (59.0–61.7) |
| Negative predictive value | 61.2 (60.5–61.9) | 59.9 (59.3–60.6) |
| Brier score | 27.7 (27.4–28.2) | 27.4 (27.1–27.8) |

CI: Confidence interval

Model 1 includes 11 maternal and perinatal characteristics.

Model 2 includes 32 maternal, obstetrical, and perinatal characteristics.

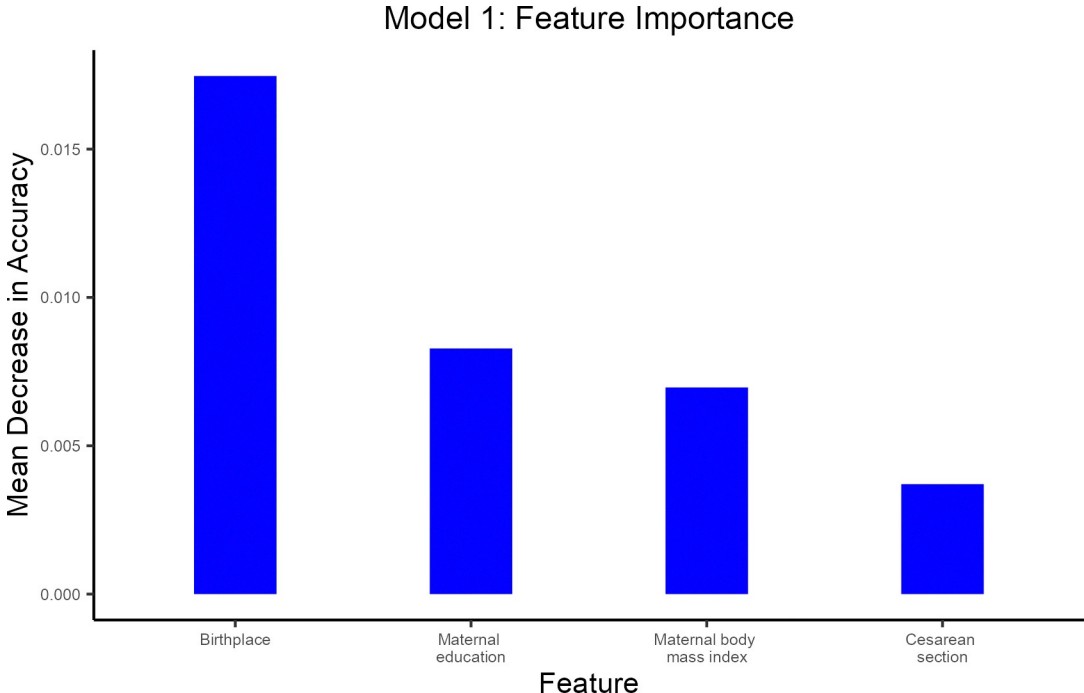

**Fig 3. Feature importance plot for model 1 to predict cessation of exclusive breastfeeding within one month.** Model 1 includes 11 maternal and perinatal characteristics (maternal age, maternal pre-pregnancy body mass index, maternal smoking, maternal education, birthplace, parity, multiple birth, delivery mode, infant's sex, gestational age, and birthweight). The sequential rank agreement was stable across the 4 highest ranking predictors.

interpretability [44]. We did not compare multiple machine learning algorithms, as the scope of the study was to investigate whether the dataset could be employed to predict exclusive breastfeeding cessation within one month. It is highly unlikely that other algorithms would yield models with a clinically relevant increase in performance.

The main limitation of the study is the potential misclassification of cessation of exclusive breastfeeding within one month. The structure of The Danish National Child Health Register entails that mother-infant pairs who did not initiate exclusive breastfeeding not were included in the register [29]. Accordingly, we classified mother-infant pairs without a record in the register as having ceased exclusive breastfeeding within one month. However, other factors might result in missing records, e.g., rejection of health visitor services or errors in reporting to the register. More than 95% of Danish parents use health visitor services, but we cannot exclude that there might be slight deviations from the recommended five visits in the infant's first year [8].

We applied data from 2014 and 2015 to develop the models due to the data validity in these years. It is important to consider the possibility that data could be outdated and thereby affect the predictive performance in the present. We consider it unlikely that data should be outdated, as there have been no substantial changes in obstetric care, neonatal care, or breastfeeding support in Denmark during this period. However, it would be important to consider this in case of an implementation phase.

We expected that the high number of study participants and inclusion of potentially important predictors would enable us to create a valuable prediction model to select mother-infant pairs for targeted breastfeeding supportive interventions. We consider that our results corroborate that breastfeeding success depend on factors that were not encompassed in our dataset.

## Model 2: Feature Importance

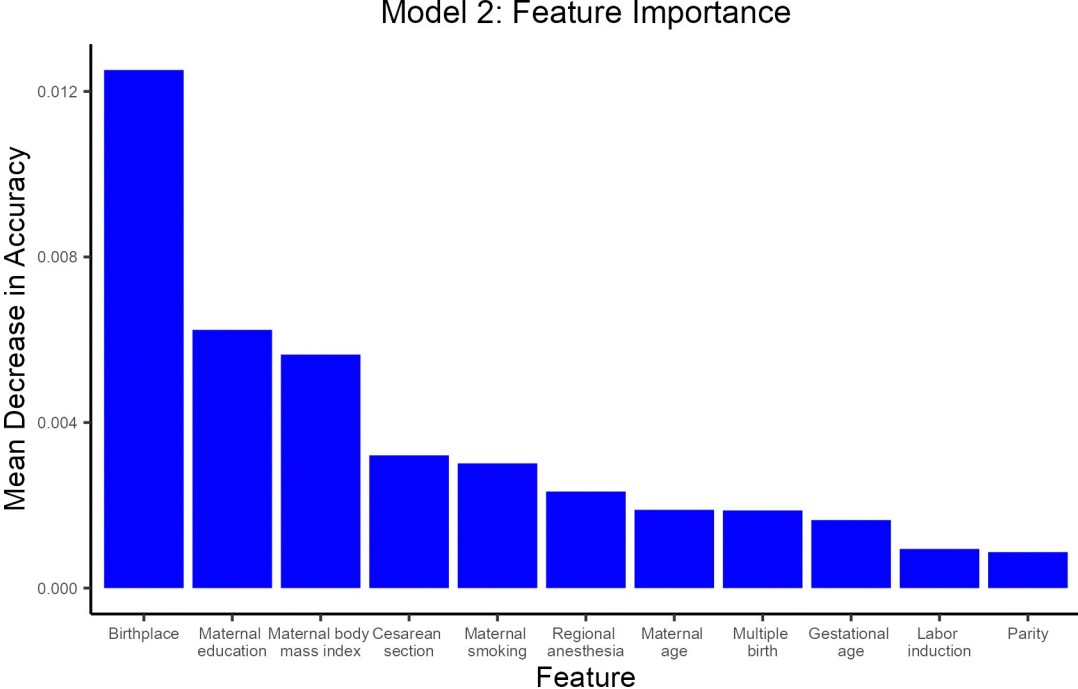

**Fig 4. Feature importance plot for model 2 to predict cessation of exclusive breastfeeding within one month.** Model 2 includes 32 maternal, obstetrical, and perinatal characteristics (maternal age, maternal pre-pregnancy body mass index, maternal smoking, maternal education, birthplace, parity, multiple birth, delivery mode, infant's sex, gestational age, birthweight, ethnicity, maternal psychiatric disease, maternal somatic chronic disease, preeclampsia and eclampsia, hemorrhage in early pregnancy, gestational diabetes mellitus, liver disease, hemorrhage in late pregnancy, preterm premature rupture of membranes, placenta previa, abruptio placenta, abnormal forces of labor, uterine rupture, postpartum hemorrhage, retention of placenta or membranes, perineal tear, labor induction, forceps or vacuum extraction, regional anesthesia, general anesthesia, and Apgar score). The sequential rank agreement was stable across the 11 highest ranking predictors.

This is a specific limitation of our method and a general limitation of machine learning models used to predict specific outcomes. The register-based design invokes general limitations including data being pre-collected by others [41], which confines us to the predictors that are available in the registers. Our two models both predicted cessation of exclusive breastfeeding with areas under the receiver operating curves on approximately 62%. To our knowledge, the best performing models made in previous studies have predicted exclusive breastfeeding with AUCs between 74% and 78% [16, 17, 20]. In addition to socio-demography, pregnancy and birth-related data, these models comprise information on breastfeeding practices, e.g., breastfeeding intention, previous breastfeeding experience, breastfeeding education, Baby-friendly Hospital Initiative designation, skin-to-skin contact between mother and infant, early breastfeeding initiation, and maternal self-efficacy. In Denmark, nearly all expectant mothers intend to breastfeed, and no hospitals hold a valid Baby-friendly Hospital Initiative designation [4]. Thus, we speculate that data on previous breastfeeding experience, breastfeeding education, skin-to-skin contact between mother and infant, early breastfeeding initiation, and maternal self-efficacy would improve the predictive performance of our models.

Our study verified well-established predictors of breastfeeding. In both models, the most important predictors were birthplace, maternal body mass index, maternal smoking, and cesarean section. Multiple studies have shown that maternal body mass index, maternal smoking, and cesarean section are associated with exclusive breastfeeding [9, 12]. Denmark is divided in five health regions. We were surprised that there were major differences between

these. We did not explore this because we focused the analyses on factors that could be generalized to other countries. We speculate that birthplace cover multiple aspects including socioeconomic status, local approach to breastfeeding, and completeness of data reported to The Danish National Child Health Register.

Breastfeeding supportive interventions, e.g., parental breastfeeding education and training of nurses, can increase exclusive breastfeeding rates [45, 46]. Different strategies have been developed to target such interventions including stratification by parity. In Denmark, breastfeeding support is initiated in the hospital. Preterm and sick infants are admitted to the neonatal ward, while healthy infants are discharged directly from the delivery ward (only multiparous) or admitted to the maternity ward. Breastfeeding support continues after discharge guided by health nurses conducting free home visits. To target support interventions on an individual level would require more accurate predictions than the models we have presented can provide. If the models had demonstrated better performance, they could be applied in the hospital after birth to identify mother-infant pairs susceptible to not establishing exclusive breastfeeding. This approach would enable targeted support interventions. In the Danish system, they could include additional home visits by health visitors or additional appointments at the hospital to promote breastfeeding establishment. To evaluate the effect of this approach, the Plan-Do-Study-Act cycle could be employed [47]. We would begin with a localized implementation of the model and targeted interventions. Should the results prove positive, we would consider expanding the initiative nationally.

Contrary to our expectations, the predictive performance was not increased by inclusion of additional predictors associated with complications during pregnancy and birth. It is possible that some of the additional predictors could provide additive value in predicting other breastfeeding outcomes. It is important to underline that prediction models cannot meaningfully be used to infer anything about biology. While complications during pregnancy and birth did not increase the predictive performance it remains possible that these factors impede exclusive breastfeeding.

## Conclusion

The two models developed did not accurately predict cessation of exclusive breastfeeding within one month among infants born after 35 weeks gestation. Contrary to our expectations, including additional factors in the model did not increase model performance. These findings underscore the complexity of predicting breastfeeding outcomes and emphasizes the need for further research to target breastfeeding supportive interventions.

## Supporting information

**S1 Table. Definitions of obstetric outcomes based on International Classification of Diseases 10th revision and NOMESCO Classification of Surgical Procedure codes in The Danish National Patient Register.**
(TIF)

**S2 Table. Definition of maternal chronic somatic disease based on International Classification of Diseases 10th revision codes in The Danish National Patient Register.**
(TIF)

**S3 Table. Definition of maternal psychiatric disease based on International Classification of Diseases 10th revision codes in The Danish National Patient Register.**
(TIF)

**S4 Table. Performance of the models' prediction of exclusive breastfeeding cessation within one month in the development data.**
(TIF)

**S1 Fig. Sequential rank agreement for the feature importance analyses in model 1 to predict cessation of exclusive breastfeeding within one month.**
(TIF)

**S2 Fig. Sequential rank agreement for the feature importance analyses in model 2 to predict cessation of exclusive breastfeeding within one month.**
(TIF)

## Author Contributions

**Conceptualization:** Freja Marie Nejsum, Rikke Wiingreen, Andreas Kryger Jensen, Ellen Christine Leth Løkkegaard, Bo Mølholm Hansen.

**Data curation:** Freja Marie Nejsum, Rikke Wiingreen.

**Formal analysis:** Freja Marie Nejsum, Rikke Wiingreen, Andreas Kryger Jensen, Ellen Christine Leth Løkkegaard, Bo Mølholm Hansen.

**Investigation:** Rikke Wiingreen, Andreas Kryger Jensen, Ellen Christine Leth Løkkegaard, Bo Mølholm Hansen.

**Methodology:** Freja Marie Nejsum, Rikke Wiingreen, Andreas Kryger Jensen, Ellen Christine Leth Løkkegaard, Bo Mølholm Hansen.

**Project administration:** Freja Marie Nejsum, Rikke Wiingreen, Bo Mølholm Hansen.

**Resources:** Rikke Wiingreen, Bo Mølholm Hansen.

**Software:** Freja Marie Nejsum, Rikke Wiingreen, Andreas Kryger Jensen.

**Supervision:** Rikke Wiingreen, Andreas Kryger Jensen, Ellen Christine Leth Løkkegaard, Bo Mølholm Hansen.

**Validation:** Rikke Wiingreen, Andreas Kryger Jensen, Ellen Christine Leth Løkkegaard, Bo Mølholm Hansen.

**Visualization:** Freja Marie Nejsum, Rikke Wiingreen, Andreas Kryger Jensen, Ellen Christine Leth Løkkegaard, Bo Mølholm Hansen.

**Writing – original draft:** Freja Marie Nejsum, Bo Mølholm Hansen.

**Writing – review & editing:** Freja Marie Nejsum, Rikke Wiingreen, Andreas Kryger Jensen, Ellen Christine Leth Løkkegaard, Bo Mølholm Hansen.

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
