## [Decision Letter · Decision Letter 0]

31 Jul 2024

PONE-D-24-15143Predicting early cessation of exclusive breastfeeding using machine learning techniquesPLOS ONE

Dear Dr. Nejsum,

Thank you for submitting your manuscript to PLOS ONE. After careful consideration, we feel that it has merit but does not fully meet PLOS ONE’s publication criteria as it currently stands. Therefore, we invite you to submit a revised version of the manuscript that addresses the points raised during the review process.

We look forward to receiving your revised manuscript.

Kind regards,

Astrid M. Kamperman

Academic Editor

PLOS ONE

3. For studies involving third-party data, we encourage authors to share any data specific to their analyses that they can legally distribute. PLOS recognizes, however, that authors may be using third-party data they do not have the rights to share. When third-party data cannot be publicly shared, authors must provide all information necessary for interested researchers to apply to gain access to the data. (https://journals.plos.org/plosone/s/data-availability#loc-acceptable-data-access-restrictions)

a) A description of the data set and the third-party source

b) If applicable, verification of permission to use the data set

c) Confirmation of whether the authors received any special privileges in accessing the data that other researchers would not have

d) All necessary contact information others would need to apply to gain access to the data

Reviewers' comments:

Reviewer's Responses to Questions

**Comments to the Author**

1. Is the manuscript technically sound, and do the data support the conclusions?

Reviewer #1: Yes

Reviewer #2: Yes

2. Has the statistical analysis been performed appropriately and rigorously? 

Reviewer #1: No

Reviewer #2: Yes

3. Have the authors made all data underlying the findings in their manuscript fully available?

Reviewer #1: No

Reviewer #2: No

4. Is the manuscript presented in an intelligible fashion and written in standard English?

Reviewer #1: Yes

Reviewer #2: Yes

5. Review Comments to the Author

Reviewer #1: The authors present results from a predictive model trained on a very impressive dataset. I point out some directions in which the manuscript and the methodology used within could benefit from a revisit.

1. The authors talk in the introduction about statistics and guidelines from the WHO that mention 6-months cessation. The outcome of interest in the current study is cessation after 1 month. Are the mothers that stop breast feeding less than a month after delivery the same as those that stop just under 6 months? Both would technically fall under the 6 month recommendation, yet intuitively I would think they would be very different groups of people with very different reasons for stopping. Could the authors provide further information to bridge their research question of 1 month cessation to the literature and guidelines of 6 month cessation?

2. Why does the observational period cover 2 years (2014 – 2015)? Naturally, adding more years would produce a sample so big that computation time might become an inconvenience. That is fine, but could the authors add a description of their rationale for this cut-off

3. Relatedly, instead of validating the models on folds from 2014/2015, it would be a lot more convincing to test how accurate predictions are on later years. That would be strong evidence the models generalize beyond the time frame under study.

4. Later in the methods, the authors state they use cross-validation to check the final performance of the model. It does not seem the authors held out part of the data exclusively for testing? It’s okay to use the same data to train and test models using cross-validation when tuning hyperparameters. For the final test, however, there should be a holdout dataset that has not been used at all in the training process. This is the only fair test of performance. See the point above for an idea to use later year(s) as the testing data.

5. Why did the authors choose random forest specifically instead of testing several algorithms against each other? There are alternatives that could generate improvements in prediction, including a boosted random forest for example or an ensemble of several models even

6. The stated aim of the paper is to create prediction models. Yet an algorithm that creates causal DAGs is used. The figure presented is visually striking, yet I would not include it in this manuscript. Causal inference requires careful consideration of which variables to include in a model and the structural relationships between these variables. The amount of variables included in the current paper are appropriate for prediction, but for causal inference they are likely to produce very biased associations. I would stick to prediction for these results

7. I would not report on p-values in Table 1. The huge sample size guarantees that trivially small differences will result in significant differences, making p-values practically uninformative.

8. Recent guidelines on publishing predictive models require some information on how the model would be implemented in the real world. Who would be doing the monitoring, how would the results be communicated and to whom? I would devote some text either in the introduction or the discussion thinking through the hows and whys of implementing this model regardless of whether it successfully predicts or not.

Reviewer #2: Major points

1. What are the potential risks of using outdated data (2014 and 2015)?

2. An explanation for the choice of the 35-week gestational age cutoff should be provided.

3. For the calculus of the birth weight outliers, it would be useful if the authors provided the mean and standard deviation of birth weight in this study (lines 96-98).

4. The study criteria to construct the outcome considered the use of a maximum of one formula feeding per week after hospital discharge (lines 101-103). The authors could report the numbers of infants who were exclusively breastfed, considering the groups with and without formula feeding.

5. What is the periodicity of free home visits in the first month of a child?

6. I suggested that the authors explain how many children did not have a record of cessation of exclusive breastfeeding within the first month after birth (lines 111-113), and if it includes those who did not initiate exclusive breastfeeding.

7. The birthplace is an important variable in the findings, but the healthcare regions A-E are not completely clear to a reader from the other countries. It is suggested that the authors characterize these regions in the methods section, mainly by the socioeconomic aspects.

8. Since the authors use a multiple imputation method, what was the percentage of missing data?

9. It is suggested that authors add a reference or more explanation about Rubin’s rule (line 150) and PC-algorithm (e.g. Peter-Clark algorithm) (line 154).

10. Authors mention that the lower sparsity levels result in the strongest pathways. However, the methods section did not include an explanation of how the sparsity levels are chosen (lines 157-159). Maybe the sentence in the results section (“Since there is no golden standard method to determine the optimal sparsity level, we chose … three sparsity levels”, lines 217-218) could be moved to the methods section.

11. It is suggested that the authors add subtitles in the methods section according to the statistical analysis step (e.g. causal analysis, prediction, and validation) to help the reader.

12. Table 1 provided p-values, but the statistical test and significance level are not included in the methods section.

13. The text about Table 1 (Table 1 shows…one month, lines 205-207) only cited some associations. How are criteria used to consider the differences between the groups with and without cessation of exclusive breastfeeding? Moreover, it is important to note that p-values can be less informative in decision-making with big data, especially due to the large sample size.

14. The authors used three sparcity levels in the results section, but they chose the 10-185 level. What is the difference between them according to the DAGs and the relation of variables with breastfeeding cessation, and why does it seem the better sparcity level?

15. The explanation about the multiple imputation is before the prediction models, but if I understand it is used during prediction modeling.

16. The feature importance could be described in the methods section in the same order as the results section.

17. The authors could explore the reasons why other studies had higher AUCs, by looking beyond breastfeeding practices. It is possible that differences in the study population characteristics justified these findings (lines 314-316).

18. Although the authors did not find an important role of additional factors (21 variables) in early cessation, they could discuss the importance of these factors in other outcomes, like prolonged exclusive breastfeeding (e.g. six months).

Minor points

1. Maybe the sentence “The first model included 11 well-established risk factors for cessation of... during pregnancy and delivery that potentially impede breastfeeding.” (lines 73-76) is only necessary in the methods section.

2. The first sentence of the methods section is very similar to the end of the introduction section (“In a retrospective cohort, we applied techniques from machine learning to develop and validate”, lines 79-81). I suggest that the authors specify the local and period of data in this sentence and omit the name of the analysis method (machine learning).

3. It is recommended that authors include a reference for ISCED 2011.

4. The references for the causalDisco package and the cross-validation are not shown.

5. The percentage of missing perinatal data is not 2.7% (line 196).

6. The captions could be more specific (say children) rather than use terms like study population, beyond specifying the years of study.

7. The results section has many tables and figures. The authors could reconsider their inclusions, such as Table 2 in the manuscript.

8. The resolution of the text inside the Figures is not clear.

6. PLOS authors have the option to publish the peer review history of their article (what does this mean?). If published, this will include your full peer review and any attached files.

Reviewer #1: No

Reviewer #2: No

---

## [Author Response · Author response to Decision Letter 0]

13 Sep 2024

Thank you very much for the insightful comments. They are highly appreciated. 

Reviewer #1: The authors present results from a predictive model trained on a very impressive dataset. I point out some directions in which the manuscript and the methodology used within could benefit from a revisit.

1. The authors talk in the introduction about statistics and guidelines from the WHO that mention 6-months cessation. The outcome of interest in the current study is cessation after 1 month. Are the mothers that stop breast feeding less than a month after delivery the same as those that stop just under 6 months? Both would technically fall under the 6 month recommendation, yet intuitively I would think they would be very different groups of people with very different reasons for stopping. Could the authors provide further information to bridge their research question of 1 month cessation to the literature and guidelines of 6 month cessation?

We have provided further information in the introduction to bridge the research question concerning exclusive breastfeeding cessation within one month with the literature and guidelines concerning exclusive breastfeeding cessation within six months: “The first step to achieve exclusive breastfeeding for six months after birth is to establish exclusive breastfeeding.” (lines 47-48 in Revised manuscript with track changes). 

The rationale for evaluating exclusive breastfeeding cessation within one month as an indicator for exclusive breastfeeding establishment has further been elaborated in the methods: “The outcome was cessation of exclusive breastfeeding within one month, as exclusive breastfeeding usually is well-established at this point and infants born at gestational ages above 34 weeks and 6 days routinely are discharged from the hospital beforehand” (lines 106-108 in Revised manuscript with track changes).

2. Why does the observational period cover 2 years (2014 – 2015)? Naturally, adding more years would produce a sample so big that computation time might become an inconvenience. That is fine, but could the authors add a description of their rationale for this cut-off.

We have added a paragraph explaining the rationale for the two year cut-off in the methods: “This practice has been mandatory since 2011 but data are only considered complete from the 1st of January 2014 [1]. The reporting is conducted via municipalities, which are local administrative divisions responsible for public services within specific geographic areas of Denmark. This leads to considerable delay in reporting of data to The Danish National Child Register. Further, post-registrations dating several years back are possible, thus data on can only be considered complete after multiple years [2].” (lines 124-130 in Revised manuscript with track changes).

3. Relatedly, instead of validating the models on folds from 2014/2015, it would be a lot more convincing to test how accurate predictions are on later years. That would be strong evidence the models generalize beyond the time frame under study.

We agree that validating the model on data from later years would be ideal. However, we have concerns for the data validity in later years because of how data are reported to The Danish National Child Register. We have provided further information on how data are reported to The Danish National Child Register: “The reporting is conducted via municipalities, which are local administrative divisions responsible for public services within specific geographic areas of Denmark. This leads to considerable delay in reporting of data to The Danish National Child Register. Further, post-registrations dating several years back are possible, thus data on can only be considered complete after multiple years [2].” (lines 125-130 in Revised manuscript with track changes). These concerns are further substantiated because the exclusive breastfeeding rates reported in The Danish National Child Register have decreased in later years. We consider this unlikely given that there have not been significant changes in breastfeeding support in Denmark during this period. 

4. Later in the methods, the authors state they use cross-validation to check the final performance of the model. It does not seem the authors held out part of the data exclusively for testing? It’s okay to use the same data to train and test models using cross-validation when tuning hyperparameters. For the final test, however, there should be a holdout dataset that has not been used at all in the training process. This is the only fair test of performance. See the point above for an idea to use later year(s) as the testing data.

Thank you for pointing this out. Inspired by your comments, we chose to withhold part of the data for validation. We have made the following revisions in the manuscript to reflect this approach:

• In the methods, we have added a paragraph that explain how data were divided into two datasets for model development and model validation: “The data were divided into one dataset for model development and one dataset for model validation based on the infants’ birth month to prevent bias from time-related changes. The dataset for model development comprised all infants born in January, February, March, May, June, July, September, October, and November. The dataset for model validation comprised all infants born in April, August, and December.” (lines 188-192 in Revised manuscript with track changes). 

• In the methods, we have made the following adjustments in the paragraph on model development: “The models were trained on the dataset for model development. To train the two models, we applied ten-fold cross validation using the R-package ‘caret’ [3]. In ten-fold cross validation, the dataset is divided into ten equally sized folds. The model is iteratively trained and tested ten times. During each iteration, one distinct fold is used as the test set, while the remaining nine folds serve as the training set. This ensures that each data point is used for both training and testing, thereby enhancing the precision of the performance estimation [4]. We employed multiple metrics to evaluate the performance of the models including the area under the receiver operating curve (AUC), the area under the precision-recall curve, accuracy, sensitivity, specificity, positive predictive value, negative predictive value, and the Brier score.” (lines 201-209 in Revised manuscript with track changes).

• In the methods, we have made the following adjustment in the paragraph on model validation: “The performance of the two models were evaluated using the dataset for model validation.” (line 219 in Revised manuscript with track changes).

• In the results, we have described the number of infants in the datasets for model development and model validation: “The dataset for model development comprised 83,385 infants, while the dataset for model validation comprised 26,821 infants.” (lines 278-279 in Revised manuscript with track changes).

• In the results, ‘Table 2: Confusion matrix for the models’ prediction of exclusive breastfeeding cessation within one month calculated as the mean across the ten folds of test data’ has been modified to contain the confusion matrix for the model’s prediction of exclusive breastfeeding within one month in the dataset for model validation (‘Table 2: Confusion matrix for the models’ prediction of exclusive breastfeeding cessation within one month in the dataset for model validation’ in Revised manuscript with track changes).

• In the results, ‘Figure 3: The receiver operating curves for the two models to predict cessation of exclusive breastfeeding within one month.’ has been modified to contain the receiver operating curves for the models’ prediction of exclusive breastfeeding cessation in the dataset for model validation (‘Figure 2: The receiver operating curves for the models’ prediction of exclusive breastfeeding cessation within one month in the dataset for model validation.’ in Revised manuscript with track changes). 

• In the results, ‘Table 3: Peformance of the models to predict cessation of exclusive breastfeeding within one month’ has been modified to contain the performance parameters for the models’ prediction of exclusive breastfeeding cessation in the dataset for model validation (‘Table 3 Performance of the models’ prediction of exclusive breastfeeding cessation within one month in the dataset for model validation.’ in Revised manuscript with track changes). 

• We have added a supplementary table with the models’ performance in predicting cessation of exclusive breastfeeding within one month in the dataset for model development, evaluated using tenfold cross validation (‘S4 Table. Performance of the models’ prediction of exclusive breastfeeding cessation within one month in the data for model development.’). This information is valuable, as it provides insights into the models’ behaviors during training and thereby clarify whether the models’ limitations in the dataset for model validation are due to overfitting or inherent predictive challenges. 

• In the results, ‘Figure 4: Feature importance plot for model 1 to predict cessation of exclusive breastfeeding within one month.’ has been modified so it is now based on the dataset for model development (‘Figure 3: Feature importance plot for model 1 to predict cessation of exclusive breastfeeding within one month’ in Revised manuscript with track changes). 

• In the results, ‘Figure 5: Feature importance plot for model 2 to predict cessation of exclusive breastfeeding within one month.’ has been modified so it is now based on the dataset for model development (‘Figure 4: Feature importance plot for model 2 to predict cessation of exclusive breastfeeding within one month’ in Revised manuscript with track changes). 

5. Why did the authors choose random forest specifically instead of testing several algorithms against each other? There are alternatives that could generate improvements in prediction, including a boosted random forest for example or an ensemble of several models even.

We have added a paragraph on the rationale for choosing the random forest algorithm in the discussion: “We chose the random forest algorithm because of its widespread acceptance as a traditional machine learning method, recognized for its flexibility and interpretability [5]. We did not compare multiple machine learning algorithms, as the scope of the study was to investigate whether the dataset could be employed to predict exclusive breastfeeding cessation within one month. It is highly unlikely that other algorithms would yield models with a clinically relevant increase in performance.” (lines 361-366 in Revised manuscript with track changes). 

6. The stated aim of the paper is to create prediction models. Yet an algorithm that creates causal DAGs is used. The figure presented is visually striking, yet I would not include it in this manuscript. Causal inference requires careful consideration of which variables to include in a model and the structural relationships between these variables. The amount of variables included in the current paper are appropriate for prediction, but for causal inference they are likely to produce very biased associations. I would stick to prediction for these results

Thank you for pointing this out. After careful consideration we have chosen to exclude the causal directed acyclic graphs. 

7. I would not report on p-values in Table 1. The huge sample size guarantees that trivially small differences will result in significant differences, making p-values practically uninformative.

This has been changed.

8. Recent guidelines on publishing predictive models require some information on how the model would be implemented in the real world. Who would be doing the monitoring, how would the results be communicated and to whom? I would devote some text either in the introduction or the discussion thinking through the hows and whys of implementing this model regardless of whether it successfully predicts or not.

Thank you, that is a good point. We have added a paragraph on implementation of the model in the discussion: “If the models had demonstrated better performance, they could be applied in the hospital after birth to identify mother-infant pairs susceptible to not establishing exclusive breastfeeding. This approach would enable targeted support interventions. In the Danish system, they could include additional home visits by health visitors or additional appointments at the hospital to promote breastfeeding establishment. To evaluate the effect of this approach, the Plan-Do-Study-Act cycle could be employed [6]. We would begin with a localized implementation of the model and targeted interventions. Should the results prove positive, we would consider expanding the initiative nationally.” (lines 417-424 in Revised manuscript with track changes). 

We have further clarified the intended use of the models in the introduction: “With this study, we aimed to develop and validate two models to predict cessation of exclusive breastfeeding within one month among infants born after 35 weeks gestation using machine learning techniques, with potential for application in the hospital immediately after birth to target support interventions.” (Lines 73-76 in Revised manuscript with track changes).

Reviewer #2: Major points

1. What are the potential risks of using outdated data (2014 and 2015)?

We have added a paragraph on the potential risks of using outdated data in the discussion: “We applied data from 2014 and 2015 to develop the models, due to the data validity in these years. It is important to consider the possibility that data could be outdated and thereby affect the predictive performance in the present. We consider it unlikely that data should be outdated, as there have been no substantial changes in obstetric care, neonatal care, or breastfeeding support in Denmark during this period. However, it would be important to consider this in case of an implementation phase.” (lines 375-380 in Revised manuscript with track changes). 

2. An explanation for the choice of the 35-week gestational age cutoff should be provided.

We have provided an explanation for the cutoff in the methods: “The 35 weeks cutoff was chosen because, in Denmark, most infants born at gestational ages below 35 weeks and 0 days routinely are admitted to neonatal wards where they receive additional support to establish breastfeeding.” (lines 98-100 in Revised manuscript with track changes).

3. For the calculus of the birth weight outliers, it would be useful if the authors provided the mean and standard deviation of birth weight in this study (lines 96-98).

We have provided the mean and standard deviation of the birth weight in ‘Table 1. Baseline characteristics of the study population of infants born in 2014 and 2015 based on data from registers held by Statistics Denmark.’ in the results. 

4. The study criteria to construct the outcome considered the use of a maximum of one formula feeding per week after hospital discharge (lines 101-103). The authors could report the numbers of infants who were exclusively breastfed, considering the groups with and without formula feeding.

The Danish National Child Register uses the definition of exclusive breastfeeding as feeding the infant solely with breast milk except for water and maximum one formula feeding per week after hospital discharge. It is a limitation in the dataset, as we can, thus, not distinguish those who only received breast milk from those who received up to one formula feeding per week after hospital discharge. This has been clarified in the methods: “In the Danish National Child Register, exclusive breastfeeding is defined as feeding the infant solely with breast milk except for water and maximum one formula feeding per week after hospital discharge as described by The Danish Health Authority [8]. Thus, this definition was applied in the current study.” (lines 114-117 in Revised manuscript with track changes).

5. What is the periodicity of free home visits in the first month of a child?

We have added a paragraph on the periodicity of free home visits in the first month in the methods: “In the first month, health visitors conduct minimum one home visit in the first

---

## [Decision Letter · Decision Letter 1]

4 Oct 2024

Predicting early cessation of exclusive breastfeeding using machine learning techniques

PONE-D-24-15143R1

Dear Dr. Nejsum,

We’re pleased to inform you that your manuscript has been judged scientifically suitable for publication and will be formally accepted for publication once it meets all outstanding technical requirements.

Kind regards,

Astrid M. Kamperman

Academic Editor

PLOS ONE

Reviewers' comments:

Reviewer's Responses to Questions

**Comments to the Author**

1. If the authors have adequately addressed your comments raised in a previous round of review and you feel that this manuscript is now acceptable for publication, you may indicate that here to bypass the “Comments to the Author” section, enter your conflict of interest statement in the “Confidential to Editor” section, and submit your "Accept" recommendation.

Reviewer #1: All comments have been addressed

2. Is the manuscript technically sound, and do the data support the conclusions?

Reviewer #1: Yes

3. Has the statistical analysis been performed appropriately and rigorously? 

Reviewer #1: Yes

4. Have the authors made all data underlying the findings in their manuscript fully available?

Reviewer #1: No

5. Is the manuscript presented in an intelligible fashion and written in standard English?

Reviewer #1: Yes

6. Review Comments to the Author

Reviewer #1: The authors have narrowed the focus of the manuscript to its advantage and added results based on a validation dataset. The manuscript now reads substantially better and I have more trust in the metrics reported.

7. PLOS authors have the option to publish the peer review history of their article (what does this mean?). If published, this will include your full peer review and any attached files.

Reviewer #1: **Yes: **Milan Zarchev

---

## [Editor Report · Acceptance letter]

8 Nov 2024

PONE-D-24-15143R1 

PLOS ONE

Dear Dr. Nejsum, 

I'm pleased to inform you that your manuscript has been deemed suitable for publication in PLOS ONE. Congratulations! Your manuscript is now being handed over to our production team.

Kind regards, 

on behalf of

Dr. Astrid M. Kamperman 

Academic Editor

PLOS ONE